# Unveiling functions of the visual cortex using task-specific deep neural networks

**Kshitij Dwivedi**[1,2]*, **Michael F. Bonner**[3], **Radoslaw Martin Cichy**[1‡], **Gemma Roig**[2‡]*

**1** Department of Education and Psychology, Freie Universität Berlin, Germany, **2** Department of Computer Science, Goethe University, Frankfurt am Main, Germany, **3** Department of Cognitive Science, Johns Hopkins University, Baltimore, Maryland, United States of America

‡ jointly directed work.
* dwivedi@em.uni-frankfurt.de (KD); roig@cs.uni-frankfurt.de (GR)

**Data Availability Statement:** The authors confirm that all data underlying the findings are fully available without restriction. The data required to reproduce the results is available here: https://osf.

## Abstract

The human visual cortex enables visual perception through a cascade of hierarchical computations in cortical regions with distinct functionalities. Here, we introduce an AI-driven approach to discover the functional mapping of the visual cortex. We related human brain responses to scene images measured with functional MRI (fMRI) systematically to a diverse set of deep neural networks (DNNs) optimized to perform different scene perception tasks. We found a structured mapping between DNN tasks and brain regions along the ventral and dorsal visual streams. Low-level visual tasks mapped onto early brain regions, 3-dimensional scene perception tasks mapped onto the dorsal stream, and semantic tasks mapped onto the ventral stream. This mapping was of high fidelity, with more than 60% of the explainable variance in nine key regions being explained. Together, our results provide a novel functional mapping of the human visual cortex and demonstrate the power of the computational approach.

## Author summary

Human visual perception is a complex cognitive feat known to be mediated by distinct cortical regions of the brain. However, the exact function of these regions remains unknown, and thus it remains unclear how those regions together orchestrate visual perception. Here, we apply an AI-driven brain mapping approach to reveal visual brain function. This approach integrates multiple artificial deep neural networks trained on a diverse set of functions with functional recordings of the whole human brain. Our results reveal a systematic tiling of visual cortex by mapping regions to particular functions of the deep networks. Together this constitutes a comprehensive account of the functions of the distinct cortical regions of the brain that mediate human visual perception.

## 1. Introduction

The human visual system transforms incoming light into meaningful representations that underlie perception and guide behavior. This transformation is believed to take place through

io/dj7v2/ The code is available here: https://github.
com/cvai-repo/dnn2brain-function.

**Funding:** G.R. thanks the support of the Alfons and Gertrud Kassel Foundation. R.M.C. is supported by DFG grants (CI241/1-1, CI241/3-1) and the ERC Starting Grant (ERC-2018- StG 803370). The funders had no role in study design, data collection and analysis, decision to publish, or preparation of the manuscript.

**Competing interests:** The authors have declared that no competing interests exist.

a cascade of hierarchical processes implemented in a set of brain regions along the so-called ventral and dorsal visual streams [1]. Each of these regions has been stipulated to fulfill a distinct sub-function in enabling perception [2]. However, discovering the exact nature of these functions and providing computational models that implement them has proven challenging. Recently, computational modeling using deep neural networks (DNNs) has emerged as a promising approach to model, and predict neural responses in visual regions [3–7]. These studies have provided a first functional mapping of the visual brain. However, the resulting account of visual cortex functions has remained incomplete. This is so because previous studies either explain the function of a single or few candidate regions by investigating many DNNs or explain many brain regions comparing it to a single DNN trained on one task only (usually object categorization). In contrast, for a systematic and comprehensive picture of human brain function that does justice to the richness of the functions that each of its subcomponents implements, DNNs trained on multiple tasks, i.e., functions, must be related and compared in their predictive power across the whole cortex.

Aiming for this systematic and comprehensive picture for the visual cortex we here relate brain responses across the whole visual brain to a wide set of DNNs, in which each DNN is optimized for a different visual task, and hence, performs a different function.

To reliably reveal the functions of brain regions using DNNs performing different functions, we need to ensure that only function and no other crucial factor differs between the DNNs. The parameters learned by a DNN depend on a few fundamental factors, namely, its architecture, training dataset, learning mechanism, and the function the DNN was optimized for. Therefore, in this study, we select a set of DNNs [8] that have an identical encoder architecture and are trained using the same learning mechanism and the same set of training images. Thus, the parameters learned by the encoder of the selected DNNs differ only due to their different functions.

We generate a functional map of the visual cortex by comparing the fMRI responses to scene images [9] with the activations of multiple DNNs optimized on different tasks [8] related to scene perception, e.g., scene classification, depth estimation, and edge detection. Our key result is that different regions in the brain are better explained by DNNs performing different tasks, suggesting different computational roles in these regions. In particular, we find that early regions of the visual cortex are better explained by DNNs performing low-level vision tasks, such as edge detection. Regions in the dorsal stream are better explained by DNNs performing tasks related to 3-dimensional (3D) scene perception, such as occlusion detection and surface normal prediction. Regions in the ventral stream are best explained by DNNs performing tasks related to semantics, such as scene classification. Importantly, the top-3 best predicting DNNs explain more than 60% of the explainable variance in nine ventral-temporal and dorsal-lateral visual regions, demonstrating the quantitative power and potential of our AI-driven approach for discovering fine-grained functional maps of the human brain.

## 2. Results

### 2.1 Functional map of visual cortex using multiple DNNs

Our primary goal is to generate a functional map of the visual brain in terms of the functions each of the regions implements. Our approach is to relate brain responses to activations of DNNs performing different functions. For this, we used an fMRI dataset recorded while human subjects (N = 16) viewed indoor scenes [9] and performed a categorization task; and a set of 18 DNNs [8] optimized to perform 18 different functions related to visual perception (some of the tasks can be visualized here: https://sites.google.com/view/dnn2brainfunction/home#h.u0nqne179ys2) plus an additional DNN with random weights as a baseline. The

different DNNs' functions were associated with indoor scene perception, covering a broad range of tasks from low-level visual tasks, (e.g., edge detection) to 3-dimensional visual perception tasks (e.g., surface normals prediction) to categorical tasks (e.g., scene classification). Each DNN consisted of an encoder-decoder architecture, where the encoder had an identical architecture across tasks, and the decoder varied depending on the task. To ensure that the differences in variance of fMRI responses explained by different DNNs from our set were not due to differences in architecture, we selected the activations from the last two layers of the identical encoder architecture for all DNNs.

The layer selection was based on an analysis finding the most task-specific layers of the encoder (see S1 Text and S2 Fig). Furthermore, all DNNs were optimized using the same set of training images, and the same backpropagation algorithm for learning. Hence, any differences in our findings across DNNs cannot be attributed to the training data statistics, architecture, or learning algorithm, but to the task for which each DNN was optimized.

To compare fMRI responses with DNNs, we first extracted fMRI responses in a spatially delimited portion of the brain for all images in the stimulus set (Fig 1A). This could be either a group of spatially contiguous voxels for searchlight analysis [10–12] or voxels confined to a particular brain region as defined by a brain atlas for a region-of-interest (ROI) analysis. Equivalently, we extracted activations from the encoders of each DNN for the same stimulus set.

We then used Representational Similarity Analysis (RSA) [13] to compare brain activations with DNN activations. RSA defines a similarity space as an abstraction of the incommensurable multivariate spaces of the brain and DNN activation patterns. This similarity space is defined by pairwise distances between the activation patterns of the same source space, either fMRI responses from a brain region or DNN activations, where responses can be directly related. For this, we compared all combinations of stimulus-specific activation patterns in each source space (i.e., DNN activations, fMRI activations). Then, the results for each source space were noted in a two-dimensional matrix, called representational dissimilarity matrices (RDMs). The rows and columns of RDMs represent the conditions compared. To relate fMRI and DNNs in this RDM-based similarity space we performed multiple linear regression predicting fMRI RDM from DNN RDMs of the last two encoder layers. We obtained the adjusted coefficient of determination $R^2$ (referred to as $R^2$ in the subsequent text) from the regression to quantify the similarity between the fMRI responses and the DNN (Fig 1B). We performed this analysis for each of the 18 DNNs investigated, which we group into 2D, 3D, or semantic DNNs when those are optimized for 2D, 3D, or semantic tasks, respectively, and an additional DNN with random weights as a baseline. The tasks were categorized into three groups (2D, 3D, and semantic) based on different levels of indoor scene perception and were verified in previous works using transfer performance using one DNN as the initialization to other target tasks [8] and representational similarity between DNNs [14]. We finally used the obtained DNN rankings based on $R^2$ to identify the DNNs with the highest $R^2$ for fMRI responses in that brain region (Fig 1C top). To visualize the results, we color-coded the brain region by color indexing the DNN showing the highest $R^2$ in that brain region (Fig 1C bottom).

To generate a functional map across the whole visual cortex we performed a searchlight analysis [11,12]. In detail, we obtain the $R^2$-based DNN rankings on the local activation patterns around a given voxel, as described above. We conducted the above analysis for each voxel, resulting in a spatially unbiased functional map.

We observed that different regions of the visual cortex showed the highest similarity with different DNNs. Importantly, the pattern with which different DNNs predicted brain activity best was not random but spatially organized: 2D DNNs (in shades of blue in Fig 1D; interactive map visualization available here: https://sites.google.com/view/dnn2brainfunction/home#h. ub1chq1k42n6) show a higher similarity with early visual regions, 3D DNNs (in shades of

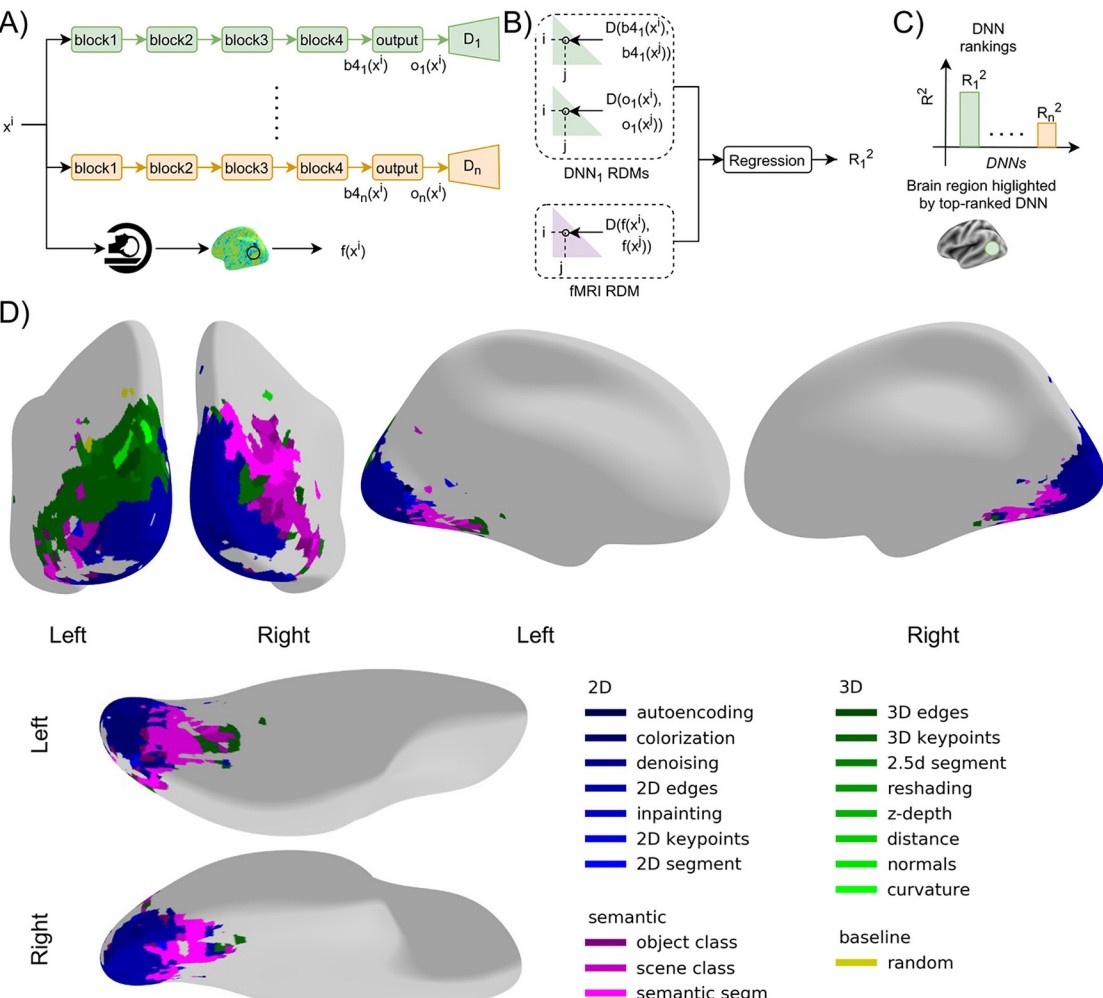

**Fig 1. Methods and results of functional mapping of the visual cortex by task-specific DNNs. A**) Schema of DNN-fMRI comparison. As a first step, we extracted DNN activations from the last two layers (block 4 and output) of the encoders, denoted as $b4_1(x^i)$, $o_1(x^i)$ for $DNN_1$ and $b4_n(x^i)$, $o_n(x^i)$ for $DNN_n$ in the figure, from n DNNs and the fMRI response of a region $f(x^i)$ for the $i^{th}$ image $x^i$ in the stimulus set. We repeated the above procedure for all the images in the stimulus set. **B**) We used the extracted activations to compute the RDMs, two for the two DNN layers and one for the brain region. Each RDM contains the pairwise dissimilarities of the DNN activations or brain region activations, respectively. We then used multiple linear regression to obtain an $R_1^2$ score to quantify the similarity between $DNN_1$ and the brain region. We repeated the same procedure using other DNNs to obtain corresponding $R^2$ **C**) We obtained a ranking based on $R^2$ to identify the DNNs with the highest $R^2$ for fMRI responses in that brain region. To visualize the results, we color-coded the brain region by the color indexing the DNN showing the highest $R^2$ in that brain region. **D**) Functional map of the visual brain generated through a spatially unbiased searchlight procedure, comparing 18 DNNs optimized for different tasks and a randomly initialized DNN as a baseline. We show the results for the voxels with significant noise ceiling and $R^2$ with DNN (p<0.05, permutation test with 10,000 iterations, FDR-corrected). An interactive visualization of the functional brain map is available in this weblink (https://sites.google.com/view/dnn2brainfunction/home#h.ub1chq1k42n6).

green) show a higher similarity with dorsal regions, while semantic DNNs (in shades of magenta) show a higher similarity with ventral regions and some dorsal regions.

Together, the results of our AI-driven mapping procedure suggest that early visual regions perform functions related to low-level vision, dorsal regions perform functions related to both 3D and semantic perception, and ventral regions perform functions related to semantic perception.

## 2.2 Nature and predictive power of the functional map

Using the searchlight results from Fig 1D, we identified the DNN that showed the highest $R^2$ for each searchlight. This poses two crucial questions that require further investigation for an in-depth understanding of the functions of brain regions. Firstly, does a single DNN prominently predict a region's response (one DNN-to-one region) or a group of DNNs together predict its response (many DNNs-to-one region)? A one-to-one mapping between DNN and a region would suggest a single functional role while a many-to-one mapping would suggest multiple functional roles of the brain region under investigation. Secondly, given that the DNNs considered in this study predict fMRI responses, how well do they predict on a quantitative scale? A high prediction accuracy would suggest that the functional mapping obtained using our analysis is accurate, while a low prediction accuracy would suggest that DNNs considered in this study are not suitable to find the function of that brain region. Although it is possible to answer the above questions for each voxel, for conciseness we consider 25 regions of interest (ROIs) tiling the visual cortex from a brain atlas [15].

To determine how accurately DNNs predict fMRI responses, we calculated the lower and upper bound of the noise ceiling for each ROI. We included ROIs (15 out of 25) with a lower noise ceiling above 0.1 and discarded other ROIs due to low signal-to-noise ratio. We show the locations of the investigated ROIs in the visual cortex in Fig 2A.

For each ROI we used RSA to compare fMRI responses (transformed into fMRI RDMs) with activations of all 18 DNNs plus a randomly initialized DNN as a baseline (transformed into DNN RDMs). This yielded one $R^2$ value for each DNN per region (see S3 Fig). We then selected the top-3 DNNs showing the highest $R^2$ and performed a variance partitioning analysis [16]. We used the top-3 DNN RDMs as the independent variable and the ROI RDM as the dependent variable to find out how much variance of ROI responses is explained uniquely by each of these DNNs while considered together with the other two DNNs. Using the variance partitioning analysis (method illustrated in S1 Fig) we were able to infer the amount of unique and shared variance between different predictors (DNN RDMs) by comparing the explained variance ($R^2$) of a DNN used alone with the explained variance when it was used with other DNNs. Variance partitioning analysis (Fig 2B) using the top-3 DNNs revealed the individual DNNs that explained the most variance uniquely for a given ROI along with the unique and shared variance explained by other DNNs. The DNN that detects edges explained significantly higher variance (p<0.05, permutation test, FDR corrected across DNNs) in ROIs in early and mid-level visual regions (V1v, V1d, V2v, V2d, V3v, and hV4) uniquely than the other two DNNs, suggesting a function related to edge detection. Semantic segmentation DNN explained significantly higher unique variance in ventral ROIs VO1 and VO2, suggesting a function related to the perceptual grouping of objects. 3D DNNs (3D Keypoints, 2.5D Segmentation, 3D edges, curvature) were best predicting DNNs for dorsal ROIs V3d and V3b suggesting their role in 3D scene understanding. A combination of 3D and semantic DNNs were best predicting DNNs for other ROIs (PHC1, PHC2, LO1, LO2, and V3a). It is crucial to note that if two DNNs from the same task group are in the top-3 best predicting DNNs for an ROI, the unique variance of ROI RDM explained by DNNs in the same group will generally be lower than by DNN not in the group. We have observed that DNNs in the same task group show a higher correlation with each other as compared to DNNs in other task groups [14]. A higher correlation between the DNNs of the same task group leads to an increase in shared variance and reduces the unique variance of the ROI RDM explained by within task group DNNs. For instance, we can observe this in PHC2 (also in PHC1, V3a), where two semantic DNNs explain less unique variance than a 3D DNN. Therefore, in such cases, we restrain from interpreting that one type of DNN is significantly better than others.

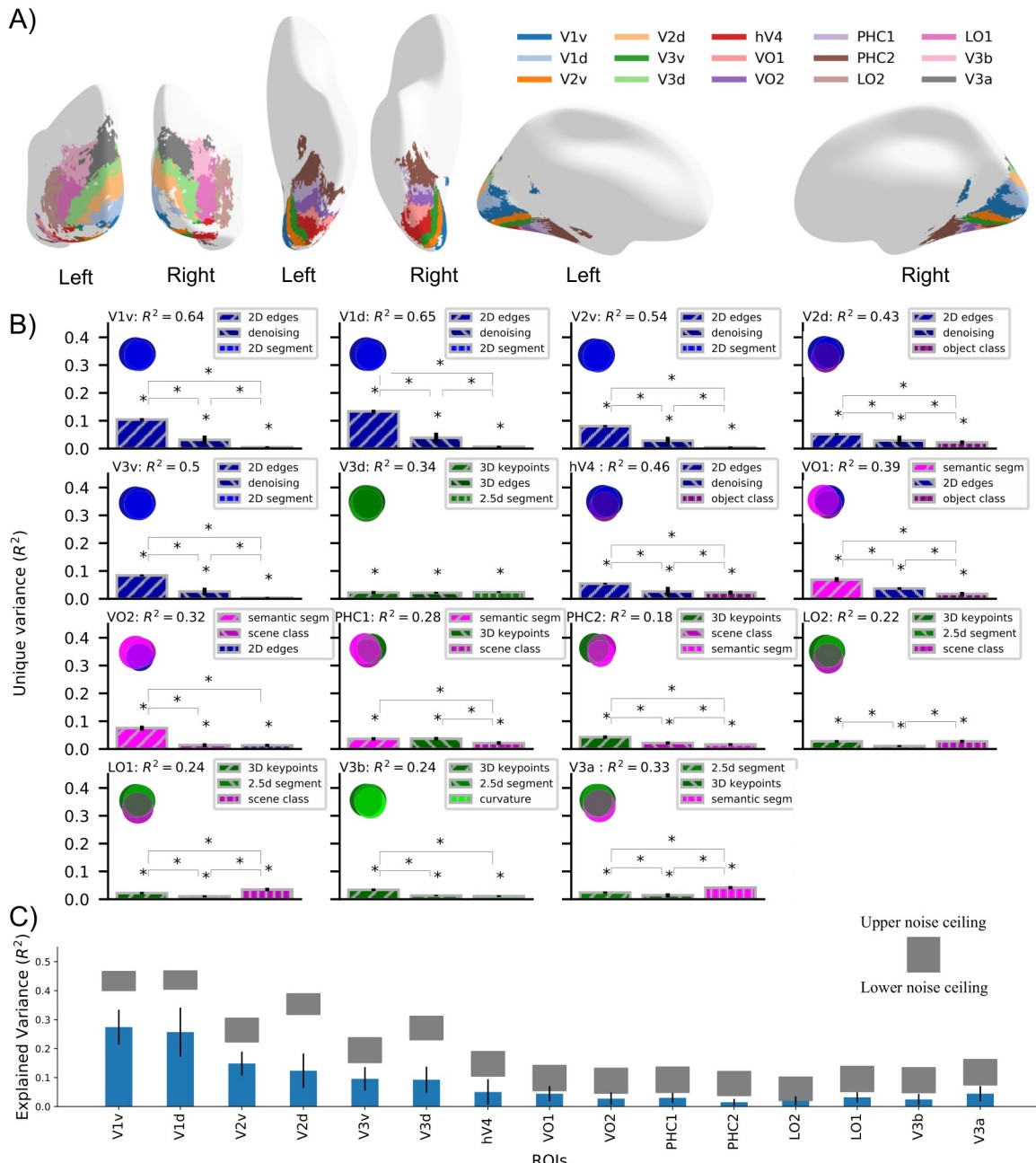

**Fig 2. Nature and predictive power of the functional map. A)** Cortical overlay showing locations of selected cortical regions from the probabilistic atlas used. **B)** Absolute total variance ($R^2$) explained in 15 ROIs by using the top-3 DNNs together. The Venn diagram for each ROI illustrates the unique and shared variance of the ROI responses explained by the combination of the top-3 DNNs. The bar plot shows the unique variance of each ROI explained by each of the top-3 DNNs individually. The asterisk denotes the significance of unique variance and the difference in unique variance ($p < 0.05$, permutation test with 10,000 iterations, FDR-corrected across DNNs). The error bars show the standard deviation calculated by bootstrapping 90% of the conditions (10,000 iterations). **C)** Variance of each ROI explained by top-3 best predicting DNNs (cross validated across subjects and conditions) indicated in blue bars compared with lower and upper bound of noise ceiling indicated by shaded gray region. The error bars show the 95% confidence interval calculated across N = 16 subjects. All the $R^2$ values are statistically significant ($p < 0.05$, two-sided t-test, FDR-corrected across ROIs).

Overall, we observed a many-to-one relationship between function and region for multiple regions, i.e., multiple DNNs explained jointly a particular brain region. In early and mid-level regions (V1v, V1d, V2v, V3v) the most predictive functions were related to low-level vision (2D edges, denoising, and 2D segmentation). In dorsal regions V3d and V3b, the most predictive functions were related to 3D scene understanding. In later ventral and dorsal regions (V2d, hV4, VO1, VO2, PHC1, PHC2, LO1, LO2, and V3a) we observed a mixed mapping of 2D, 3D, and semantic functions suggesting multiple functional roles of these ROIs. The predictability of a region's responses by multiple DNNs demonstrates that a visual region in the brain has representations well suited for distinct functions. A plausible conjecture of the above findings is that these regions might be performing a function related to the best predicting DNNs but is not present in the set of DNNs investigated in this study.

To determine the accuracy of the functional mapping of the above ROIs, we calculated the percentage of the explainable variance explained by the top-3 best predicting DNNs. We calculated the explained variance by best predicting DNNs using cross-validation across subjects (N-fold) and conditions (two-fold). As we use multiple models together for multiple linear regression, we need to cross-validate using different sets of RDMs for fitting and evaluating the fit of the regression. Here, we perform cross-validation across subjects by fitting the regression on one-subject-left-out subject-averaged RDMs on half of the images in the stimulus set and evaluating on the left-out single subject RDM on the other half of the images. The above method is a stricter evaluation criterion as compared to the commonly used one without cross-validation (See S5 Fig). We compared the variance explained by the top-3 DNNs with the lower estimate of the noise ceiling which is an estimate of the explainable variance. We found that variance explained in nine ROIs (V1v, V1d, V2v, V3v, VO1, PHC1, LO2, LO1, V3a) is higher than 60% of the lower bound of noise ceiling (Fig 2C, absolute $R^2 =$ $0.085 \pm 0.046$). In absolute terms, the minimum, median, and maximum cross-validated $R^2$ values across the 15 ROIs were 0.014 (PHC2), 0.044 (VO1), and 0.27 (V1v) which are comparable to related studies [17] performing evaluation in a similar manner. This shows that the DNNs selected in this study predict fMRI responses well and therefore are suitable for mapping the functions of the investigated ROIs.

In sum, we demonstrated that in many regions of the visual cortex, DNNs trained on different functions predicted activity. This suggests that these ROIs have multiple functional roles. We further showed quantitatively that more than 60% of the explainable variance in nine visual ROIs is explained by the set of DNNs we used, demonstrating that the selected DNNs are well suited to investigate the functional roles of these ROIs.

## 2.3 Functional map of visual cortex through 2D, 3D, and semantic tasks

In the previous section, we observed a pattern qualitatively suggesting different functional roles of early (2D), dorsal (3D and semantic), and ventral (semantic) regions in the visual cortex. To quantitatively assess this, we investigated the relation of brain responses and DNNs not at the level of single tasks, but task groups (2D, 3D, and semantic), where DNNs belonging to a task group showed a higher correlation with other DNNs in the group than with DNNs in other task groups (see S1 Text).

We averaged the RDMs of DNNs in each task group to obtain aggregate 2D, 3D, and semantic RDMs. Averaging the RDMs based on task groups reduced the number of DNN comparisons from 18 to 3. This allowed us to perform variance partitioning analysis to compare fMRI and DNN RDMs, which would be impractical with 18 single DNNs due to a large number of comparisons and computational complexity. When used in this way, variance

partitioning analysis reveals whether and where in the brain one task group explained brain responses significantly better than other task groups.

We first performed a searchlight analysis to identify where in the cortex one task group explains significantly higher variance uniquely than the other task groups. We selected the grouped DNN RDM that explains the highest variance in a given region uniquely to create a functional map of the task groups in the visual cortex (Fig 3A). Here, due to the reduced number of comparisons, we can clearly observe distinctions where one grouped DNN explains fMRI responses better than the other grouped DNNs (p<0.05, permutation test with 10,000

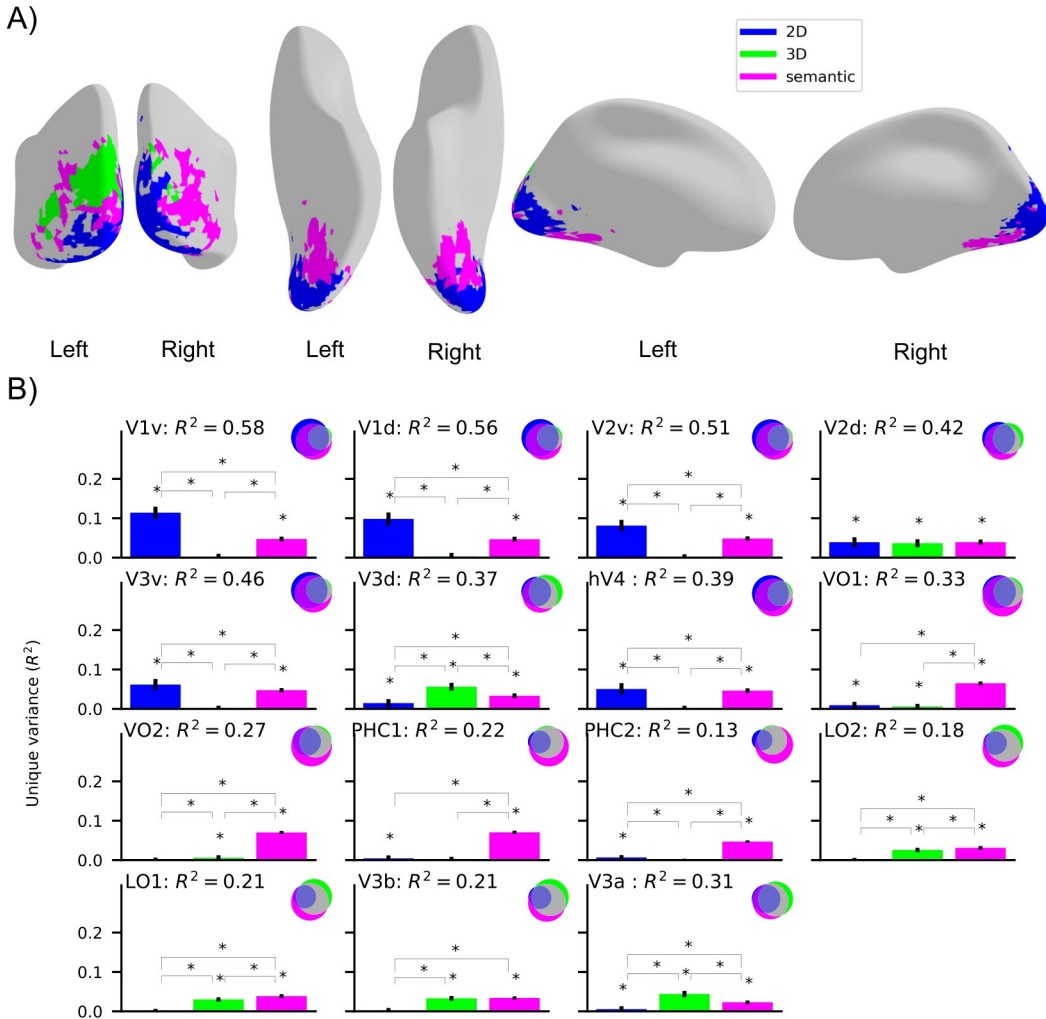

**Fig 3. Functional mapping of the visual cortex with respect to 2D, 3D, and semantic tasks. A)** Functional map of the visual cortex showing the regions where unique variance explained by one DNN group (2D, 3D, or semantic) is significantly higher than the variance explained by the other two DNN groups (p<0.05, permutation test with 10,000 iterations, FDR-corrected). We show the results for the voxels with a significant noise ceiling that show significantly higher unique variance for one DNN group than other two DNN groups (p<0.05, permutation test with 10,000 iterations, FDR-corrected across DNNs and searchlights). The functional brain map can be visualized in this weblink (https://sites.google.com/view/dnn2brainfunction/home#h.xi402x2hr0p3). **B)** Absolute variance ($R^2$) explained in 15 ROIs by using 3 DNN RDMs averaged across task groups (2D, 3D, or semantic). The Venn diagram for each ROI illustrates the unique and shared variance of the ROI responses explained by the combination of 3 task groups. The bar plot shows the unique variance of each ROI explained by each task group individually. The asterisk denotes whether the unique variance or the difference in unique variance was significant (p<0.05, permutation test with 10,000 iterations, FDR-corrected across DNNs). The error bars show the standard deviation calculated by bootstrapping 90% of the conditions (10,000 iterations).

iterations, FDR corrected across DNNs and searchlights). The resulting functional map (Fig 3A; interactive visualization available in this link: https://sites.google.com/view/dnn2brainfunction/home#h.xi402x2hr0p3) is different from the functional map in Fig 1D in two ways. First, in the functional map here we highlight the searchlight where one DNN group explained significantly higher variance uniquely than the other 2 DNN groups. In the functional map of Fig 1D, we highlighted the DNN that explained the highest variance of a searchlight without performing any statistical analysis whether the selected DNN was significantly better than the second best DNN or not due to the higher number of comparisons. Second, here we compared functions using groups of DNNs (3 functions: 2D, 3D and semantic), whereas in the previous analysis we compared functions using single DNNs (18 functions). The comparison using groups of DNNs allows us to put our findings in context with previous neuroimaging findings that are typically reported at this level.

We observed that the 2D DNN RDM explained responses in the early visual cortex, semantic DNN RDM explained responses in the ventral visual stream, and some parts in the right hemisphere of the dorsal visual stream, and 3D DNN RDM explained responses in the left hemisphere of the dorsal visual stream. The above findings quantitatively reinforce our qualitative findings from the previous section that early visual regions perform functions related to low-level vision, dorsal regions perform functions related to both 3D and semantic perception, and ventral regions perform functions related to semantic perception.

While the map of the brain reveals the most likely function of a given region, to find out whether a region can have multiple functional roles we need to visualize the variance explained by other grouped DNN RDMs along with the best predicting DNN RDM. To achieve that, we performed a variance partitioning analysis using 3 grouped DNN RDMs as the independent variable and 15 ROIs in the ventral-temporal and the dorsal-ventral stream as the dependent variable. The results in Fig 3B show the unique and shared variance explained by group-level DNN RDMs (2D, 3D, and semantic) for all the 15 ROIs.

From Fig 3B we observed that the responses in early ROIs (V1v, V1d, V2v, V3v, hV4) are explained significantly higher ($p < 0.05$, permutation test with 10,000 iterations, FDR corrected across DNNs) by 2D DNN RDM uniquely, while responses in later ventral-temporal ROIs (VO1, VO2, PHC1, and PHC2) are explained by semantic DNN RDM uniquely. In dorsal-lateral ROIs (V3a, V3d) responses are explained by 3D RDM uniquely. In LO1, LO2, and V3b 3D and semantic DNN RDMs explained significant variance uniquely while in V2d all 2D, 3D, and semantic DNN RDMs explained significant unique variance. It is crucial to note that for the ROI analysis here we use grouped DNN RDMs as compared to Fig 2B where we selected top-3 single DNNs that showed the highest $R^2$ with a given ROI. The comparison with grouped DNN RDMs provides a holistic view of the functional role of ROIs which might be missed if one of the DNNs that is related to the functional role of a ROI is not in the top-3 DNNs (as analyzed in Fig 2B). For instance, in Fig 3B the results suggest both 3D and semantic functional roles of V3b which is not evident from Fig 2B where the top 3-DNNs were all optimized on 3D tasks.

Together, we found that the functional role of the early visual cortex is related to low-level visual tasks (2D), the dorsal stream is related to tasks involved in 3-dimensional perception and categorical understanding of the scene (3D and semantic), and in the ventral stream is related to the categorical understanding of the scene (semantic).

## 2.4 Functional roles of scene-selective regions

In the previous sections, we focused on discovering functions of regions anatomically defined by an atlas. Since the stimulus set used to record fMRI responses consisted of indoor scenes, in

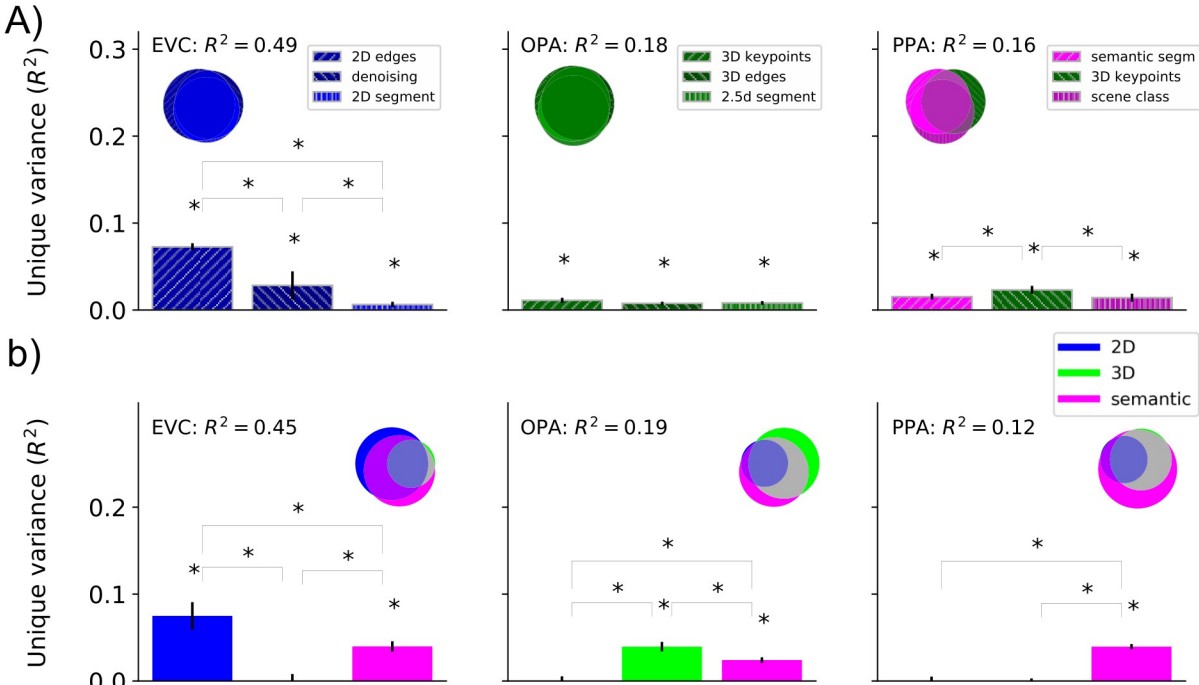

**Fig 4. Functional roles of localized ROIs. A)** Absolute total variance ($R^2$) explained in functionally localized ROIs by using the top-3 DNNs together. The Venn diagram for each ROI illustrates the unique and shared variance of the ROI responses explained by the combination of the top-3 DNNs. The bar plot shows the unique variance of each ROI explained by each of the top-3 DNNs individually. The asterisk denotes the significance of unique variance and the difference in unique variance ($p<0.05$, permutation test with 10,000 iterations, FDR-corrected across DNNs). The error bars show the standard deviation calculated by bootstrapping 90% of the conditions (10,000 iterations). **B)** Absolute total variance ($R^2$) explained in functionally localized ROIs by using 3 DNN RDMs averaged across task groups (2D, 3D, or semantic). The Venn diagram for each ROI illustrates the unique and shared variance of the ROI responses explained by the combination of 3 DNN task groups. The bar plot shows the unique variance of each ROI explained by each task group individually. The asterisk denotes whether the unique variance or the difference in unique variance was significant ($p<0.05$, permutation test with 10,000 iterations, FDR-corrected across DNNs). The error bars show the standard deviation calculated by bootstrapping 90% of the conditions (10,000 iterations).

this section we investigate functional differences in functionally localized scene-selective regions. We here focus on two major scene-selective ROIs: occipital place area (OPA) and parahippocampal place area (PPA), putting results into context with the early visual cortex (EVC) as an informative contrast region involved in basic visual processing. The analysis followed the general rationale as used before.

We first investigated the functional differences in these regions by performing variance partitioning analysis using top-3 DNNs (see $R^2$ based ranking of all DNNs in S4 Fig) that best explained a given ROIs' responses (Fig 4A). We found that the DNN that detects edges explained significantly higher variance ($p<0.05$, permutation test, FDR-corrected) in EVC uniquely than the other two DNNs, suggesting a function related to edge detection. 3D DNNs (3D Keypoints, 2.5D Segmentation, 3D edges) were best predicting DNNs for OPA suggesting its role in 3D scene understanding. A combination of semantic (semantic segmentation, scene classification) and 3D (3D keypoints) DNNs were best predicting DNNs for PPA suggesting its role in both semantic and 3D scene understanding.

We then investigated the functional differences by performing variance partitioning analysis using aggregated 2D, 3D, and semantic DNN RDMs obtained by averaging the individual DNN RDMs in each task group (Fig 4B). We found that for EVC and OPA results are highly consistent with top-3 DNN analysis showing a prominent unique variance explained by the

2D DNN RDM in EVC and the 3D DNN RDM in OPA. Interestingly, in PPA we find that the semantic DNN RDM shows the highest unique variance with no significant unique variance explained by the 3D DNN RDM. The insignificant unique variance explained by the 3D DNN RDM is potentially due to averaging the DNN RDMs of all 3D DNNs (high ranked as well as low ranked) which may lead to diminishing the contribution of an individual high ranked 3D DNN RDM (e.g. 3D keypoints that was in top-3 DNNs for PPA). Overall, we find converging evidence that OPA is mainly related to tasks involved in 3-dimensional perception (3D), and PPA is mainly related to semantic (categorical) understanding of the scene.

## 3. Discussion

In this study, we harvested the potential of discovering functions of the brain from comparison to DNNs by investigating a large set of DNNs optimized to perform a set of diverse visual tasks. We found a systematic mapping between cortical regions and function: different cortical regions were explained by DNNs performing different functions. Importantly, the selected DNNs explained 60% of the explainable variance in nine out of 15 visual ROIs investigated, demonstrating the accuracy of the AI-driven functional mapping obtained using our analysis.

Our study provides a systematic and comprehensive picture of human brain functions using DNNs trained on different tasks. Previous studies [3–7,17–24] have compared model performance in explaining brain activity, but were limited to a few preselected regions and models, or had a different goal (comparing task structure) [25]. Using the same fMRI dataset as used in this study, a previous study [18] showed that representation in scene-selective ROIs consists of both location and category information using scene-parsing DNNs. We go beyond these efforts by comparing fMRI responses across the whole visual brain using a larger set of DNNs, providing a comprehensive account of the function of human visual brain regions.

We obtained the functional mapping of different regions in the visual cortex on both individual (e.g., 2D edges, scene classification, surface normals, etc.) and group (2D, 3D, semantic) levels of visual functions. We discuss the novel insights gained at the level of individual functions that inform about the fine-grained functional role of cortical regions.

First, we consider 2D DNNs, where the denoising DNN explained significant unique variance in V1v, V1d, V2v, V2d, V3v, and hV4. The denoising task requires the DNN to reconstruct an unperturbed input image from slightly perturbed (e.g., adding Gaussian noise in the current case) input image that encourages learning representations robust to slight perturbations and limited invariance. This suggests that these ROIs might be generating a scene representation robust to high frequency noise.

When considering 3D DNNs, the 3D Keypoint and the 2.5d segment were among the top-3 best predicting DNNs in multiple ROIs. The 3D Keypoints DNN explained significant unique variance in V3d, PHC1, PHC2, LO2, LO1, V3a, V3b, OPA, and PPA. The 3D Keypoints task requires the DNN to identify locally important regions of the input image based on object boundary information and surface stability. This suggests that the ROIs in which 3D Keypoints DNN explained significant variance may be identifying locally important regions in a scene. The identification of locally important regions might be relevant to selectively attend to these key regions to achieve a behavioral goal e.g., searching for an object. The 2.5d segment DNN explained significant unique variance in V3d, LO2, LO1, V3b, V3a, and OPA. The 2.5d segment task requires the DNN to segment images into perceptually similar groups based on color and scene geometry (depth and surface normals). This suggests that the ROIs in which 2.5d segment DNN explained significant variance may be grouping regions in the images based on color and geometry cues even without any knowledge of the categorical information.

Grouping regions based on geometry could be relevant to behavioral goals such as reaching for objects or identifying obstacles.

Among semantic DNNs, the semantic segmentation DNN explained significant unique variance in VO1, VO2, PHC1, PHC2, V3a, and PPA. The semantic segmentation task requires the DNN to segment objects present in the image based on categories. This suggests that the ROIs in which semantic segmentation DNN explained significant variance may be grouping regions in the image based on categorical information.

Other DNNs (2D edges, scene classification, and object classification) that showed significant unique variance in ROIs provided functional insights mostly consistent with the previous studies [26–30]. Overall, the key DNNs (denoising, 3D keypoints, 2.5D segment, and semantic segmentation) that explained significant variance in multiple ROI responses uniquely promote further investigation by generating novel hypotheses about the functions of these ROIs. Future experiments can test these hypotheses in detail in dedicated experiments.

The functional mapping obtained using grouped DNNs is complementary to that at the individual level and helps us put functional mapping obtained here in context with previous literature. We found that early visual regions (V1v, V1d, V2v) have a functional role related to low-level 2D visual tasks which is consistent with previous literature investigating these regions [26–28]. In dorsal-ventral ROIs (V3a, V3d, LO1, and LO2) we found functional roles related to 3D and semantic tasks converging with evidence from previous studies [31–35]. Similarly, the prominent semantic functional role of later ventral-temporal ROIs (VO1, VO2, PHC1, and PHC2) found in this study converges with findings in previous literature [29–30]. In scene-selective ROIs, we found a semantic functional role for PPA and 3D functional role for OPA respectively. Our study extends the findings of a previous study [23] relating OPA and PPA to 3D models by differentiating between OPA and PPA functions through a much broader set of models. To summarize, the functional mapping using individual DNNs optimized to perform different functions revealed new functional insights for higher ROIs in the visual cortex while at the same time functional mapping using grouped DNNs showed highly converging evidence with previous independent studies investigating these ROIs.

Beyond clarifying the functional roles of multiple ROIs, our approach also identifies quantitatively highly accurate prediction models of these ROIs. We found that the DNNs explained 60% of the explainable variance in nine out of 15 ROIs. Our findings, thus, make advances towards finding models that generate new hypotheses about potential functions of brain regions as well as predicting brain responses well [21,36–38].

A major challenge in meaningfully comparing two or more DNNs is to vary only a single factor of interest while controlling the factors that may lead to updates of DNN parameters. In this study, we address this challenge by selecting a set of DNNs trained on the same set of training images using the same learning algorithm, with the same encoder architecture, while being optimized for different tasks. Our results, thus, complement previous studies that focused on other factors influencing the learning of DNN parameters such as architecture [20,39,40], and the learning mechanism [41–43]. Our approach accelerates the divide-and-conquer strategy of investigating human brain function by systematically and carefully manipulating the DNNs used to map the brain in their fundamental parameters one by one [21,44–46]. Our high-throughput exploration of potential computational functions was initially inspired by Marr's computational level of analysis [47] which aims at finding out what the goal of the computation carried out by a brain region is. While Marr's approach invites the expectation of a one-to-one mapping between regions and goals, we found evidence for multiple functional roles (3D + semantic) using DNNs in some ROIs (e.g. LO1, LO2, PHC1, PHC2). This indicates a many-to-one mapping [48] between functions and brain regions. We believe such a systematic

approach that finds the functional roles of multiple brain regions provides a starting point for a further in-depth empirical inquiry into functions of the investigated brain regions.

Our study is related to a group of studies [49–52] applying DNNs in different ways to achieve a similar goal of mapping functions of brain regions using DNNs. Some studies [49–51] applied optimization algorithms (genetic algorithm or activation maximization) to find images that maximally activate a given neuron's or group of neurons' response. Another related study [52] proposes Neural Information Flow (NIF) to investigate functions of brain regions where they train a DNN with the objective function to predict brain activity while preserving a one-to-one correspondence between DNN layers and biological neural populations. While sharing the overall goal to discover functions of brain regions, investigating DNN functions allows investigation in terms of which computational goal a given brain region is best aligned with. With new computer vision datasets [53] investigating a diverse set of tasks relevant to human behavioral goals [54,55] our approach opens new avenues to investigate brain functions.

A limitation of our study is that our findings are restricted to functions related to scene perception. Thus, the functions we discovered for non-scene regions correspond to their functions when humans are perceiving scenes. In contrast, our study does not characterize the functions of these regions when humans perceive non-scene categories such as objects, faces, or bodies. We limited our study to scene perception because there are only a few image datasets [8,56] that have annotations corresponding to a diverse set of tasks, thus, allowing DNNs to be optimized independently on these tasks. The Taskonomy dataset [8] with annotations of over 20 diverse scene perception tasks and pretrained DNNs available on these tasks along with the availability of an fMRI dataset related to scene perception [9], therefore, provided a unique opportunity. However, the approach we presented in this study is not limited to scene perception. It can in principle be extended to more complex settings such as video understanding, active visual perception, and even outside the vision modality, given an adequate set of DNNs and brain data. While in this study we considered DNNs that were trained independently, future studies might consider investigating multitask models [57,58] which are trained to perform a wide range of functions using a single DNN. Multitask modeling has the potential to model the entire visual cortex using a single model as compared to several independent models used in this study. Another potential limitation is that our findings are based on a single fMRI and image dataset, so it is not clear how well they would generalize to a broader sample of images. Given the explosive growth of the deep learning field [59] and the ever increasing availability of open brain imaging data sets [60,61] we see a furtive ground for the application of our approach in the future.

Beyond providing theoretical insight with high predictive power, our approach can also guide future research. In particular, the observed mapping between cortical region and function can serve as a quantitative baseline and starting point for an in-depth investigation focused on single cortical regions. Finally, the functional hierarchy of the visual cortex from our results can inspire the design of efficient multi-task artificial visual systems that perform multiple functions similar to the human visual cortex.

## 4. Materials and methods

### 4.1 fMRI data

We used fMRI data from a previously published study [9]. The fMRI data were collected from 16 healthy subjects (8 females, mean age 29.4 years, SD = 4.8). The subjects were scanned on a Siemens 3.0T Prisma scanner using a 64-channel head coil. Structural T1-weighted images were acquired using an MPRAGE protocol (TR = 2,200 ms, TE = 4.67 ms, flip angle = 8°,

matrix size = 192 × 256 × 160, voxel size = 0.9 × 0.9 × 1 mm). Functional T2*-weighted images were acquired using a multi-band acquisition sequence (TR = 2,000 ms for main experimental scans and 3,000 ms for localizer scans, TE = 25 ms, flip angle = 70˚, multiband factor = 3, matrix size = 96 × 96 × 81, voxel size = 2 × 2 × 2 mm).

During the fMRI scan, subjects performed a category detection task while viewing images of indoor scenes. On each trial, an image was presented on the screen at a visual angle of ~17.1˚ x 12.9˚ for 1.5 s followed by a 2.5s interstimulus interval. Subjects had to respond by pressing a button indicating whether the presented image was a bathroom or not while maintaining fixation on a cross. The stimulus set consisted of 50 images of indoor scenes (no bathrooms), and 12 control images (five bathroom images, and seven non-bathroom images). fMRI data were preprocessed using SPM12. For each participant, the functional images were realigned to the first image followed by co-registration to the structural image. Voxelwise responses to 50 experimental conditions (50 indoor images excluding control images) were estimated using a general linear model.

## 4.2 Deep neural networks

For this study, we selected 18 DNNs trained on the Taskonomy [8] dataset optimized on 18 different tasks covering different aspects of indoor scene understanding. The Taskonomy dataset is a large-scale indoor image dataset consisting of annotations for 18 single image tasks, thus, allowing optimization of DNNs on 18 different tasks using the same set of training images. We briefly describe the objective functions and DNN architectures below. For a detailed description, we refer the reader to Zamir et al. [8].

**4.2.1 Tasks and objective functions of the DNNs.**   The Taskonomy dataset consists of annotations for tasks that require pixel-level information such as edge detection, surface normal estimation, semantic segmentation, etc. as well as high-level semantic information such as object/scene classification probabilities. The tasks can be broadly categorized into 4 groups: relating to low-level visual information (2D), the three-dimensional layout of the scene (3D), high-level object and scene categorical information (semantic), and low-dimensional geometry information(geometrical). The above task categorization was obtained by analyzing the relationship between the transfer learning performance on a given task using the models pretrained on other tasks as the source tasks. The 2D tasks were edge detection, keypoint detection, 2D segmentation, inpainting, denoising, and colorization; 3D tasks were surface normals, 2.5D segmentation, occlusion edges, depth estimation, curvature estimation, and reshading; semantic tasks were object/scene classification and semantic segmentation, and low-dimensional geometric tasks were room layout estimation and vanishing point. A detailed description of all the tasks and annotations is provided in http://taskonomy.stanford.edu/taskonomy_supp_CVPR2018.pdf. In this study, we did not consider low dimensional geometric tasks as they did not fall into converging clusters according to RSA and transfer learning as in the case of 2D, 3D, and semantics tasks. To perform a given task, DNN's parameters were optimized using an objective function that minimizes the loss between the DNN prediction and corresponding ground truth annotations for that task. All the DNNs' parameters were optimized using the corresponding objective function, on the same set of training images. Due to the use of the same set of training images the learned DNN parameters vary only due to the objective function and not the difference in training dataset statistics. A complete list of objective functions used to optimize for each task is provided in this link (https://github.com/StanfordVL/taskonomy/tree/master/taskbank). We downloaded the pretrained models using this link (https://github.com/StanfordVL/taskonomy/tree/master/taskbank), where further details can be found.

**4.2.2 Network architectures.**   The DNN architecture for each task consists of an encoder and a decoder. The encoder architecture is consistent across all the tasks. The encoder architecture is a modified ResNet-50 [62] without average pooling and convolutions with stride 2 replaced by convolutions with stride 1. ResNet-50 is a 50-layer DNN with shortcut connections between layers at different depths. Consistency of encoder architecture allows us to use the outputs of the ResNet-50 encoder as the task-specific representation for a particular objective function. For all the analysis in this study, we selected the last two layers of the encoder as the task-specific representation of the DNN. Our selection criteria was based on an analysis (see S1 Text and S2 Fig) that shows task-specific representation is present in those layers as compared to earlier layers. In this way, we ensure that the difference in representations is due to the functions these DNNs were optimized for and not due to the difference in architecture or training dataset. The decoder architecture is task-dependent. For tasks that require pixel-level prediction, the decoder is a 15-layer fully convolutional model consisting of 5 convolutional layers followed by alternating convolution and transposed convolutional layers. For tasks, which require low dimensional output, the decoder consists of 2–3 fully connected layers.

## 4.3 Representational Similarity Analysis (RSA)

To compare the fMRI responses with DNN activations we first need to map both the modalities in a common representational space and then by comparing the resulting mappings we can quantify the similarity between fMRI and DNNs. We mapped the fMRI responses and DNN activations to corresponding representational dissimilarity matrices (RDMs) by computing pairwise distances between each pair of conditions. We used the variance of upper triangular fMRI RDM ($R^2$) explained by DNN RDMs as the measure to quantify the similarity between fMRI responses and DNN activations. To calculate $R^2$, we assigned DNN RDMs (RDMs of the last two layers of the encoder) as the independent variables and assigned fMRI RDM as the dependent variable. Then a multiple linear regression was fitted to predict fMRI RDM from the weighted linear combination of DNN RDMs. We evaluated the fit by estimating the variance explained ($R^2$). We describe how we mapped from fMRI responses and DNN activations to corresponding RDMs in detail below.

*Taskonomy DNN RDMs.* We selected the last two layers of the Resnet-50 encoder as the task-specific representation of DNNs optimized on each task. For a given DNN layer, we computed the Pearson's distance between the activations for each pair of conditions resulting in a condition x condition RDM for each layer. This resulted in a single RDM corresponding to each DNN layer. We followed the same procedure to create RDMs corresponding to other layers of the network. We averaged the DNN RDMs across task clusters (2D, 3D, and semantic) to create 2D, 3D, and semantic RDMs.

*Probabilistic ROI RDMs.* We downloaded probabilistic ROIs [15] from the link (http://scholar.princeton.edu/sites/default/files/napl/files/probatlas_v4.zip). We extracted activations of the probabilistic ROIs by applying the ROI masks on the whole brain response pattern for each condition, resulting in ROI-specific responses for each condition for each subject. Then for each ROI, we computed the Pearson's distance between the voxel response patterns for each pair of conditions resulting in a RDM (with rows and columns equal to the number of conditions) independently for each subject. To compare the variance of ROI RDM explained by DNN RDMs with the explainable variance we used independent subject RDMs. For all the other analyses, we averaged the RDMs across the subjects resulting in a single RDM for each ROI due to a higher signal to noise ratio in subject averaged RDMs.

*Searchlight RDMs.* We used Brainiak toolbox code [63] to extract the searchlight blocks for each condition in each subject. The searchlight block was a cube with radius = 1 and edge

size = 2. For each searchlight block, we computed the Pearson's distance between the voxel response patterns for each pair of conditions resulting in a RDM of size condition times condition independently for each subject. We then averaged the RDMs across the subjects resulting in a single RDM for each searchlight block.

## 4.4 Variance partitioning

Using RSA to compare multiple DNNs we do not obtain a complete picture of how each model is contributing to explaining the fMRI responses when considered in conjunction with other DNNs. Therefore, we determined the unique and shared contribution of individual DNN RDMs in explaining the fMRI ROI RDMs when considered with the other DNN RDMs using variance partitioning.

We performed two variance partitioning analyses on probabilistic ROIs: first using the top-3 DNNs that best explained a given ROI's responses and second using RDMs averaged according to task type (2D, 3D, and semantic). For the first analysis, we assigned a fMRI ROI RDM as the dependent variable (referred to as predictand) and assigned RDMs corresponding to the top-3 DNNs as the independent variables (referred to as predictors). For the second analysis, we assigned an fMRI ROI (searchlight) RDM as the dependent variable (referred to as predictand). We then assigned three DNN RDMs (2D, 3D, and semantic) as the independent variables (referred to as predictors).

For both variance partitioning analyses, we performed seven multiple regression analyses: one with all three independent variables as predictors, three with different pairs of two independent variables as the predictors, and three with individual independent variables as the predictors. Then, by comparing the explained variance ($R^2$) of a model used alone with the explained variance when it was used with other models, we can infer the amount of unique and shared variance between different predictors (see S1 Fig).

## 4.5 Searchlight analysis

We perform two different searchlight analyses in this study: first to find out if different regions in the brain are better explained by DNNs optimized for different tasks and second to find the pattern by taking the averaged representation DNNs from three task types (2D, 3D, and semantic). In the first searchlight analysis, we applied RSA to compute the variance of each searchlight block RDM explained by 19 DNN RDMs (18 Taskonomy DNNs and one randomly initialized as a baseline) independently. We then selected the DNN that explained the highest variance as the preference for the given searchlight block. In the second searchlight analysis, we applied variance partitioning with 2D, 3D, and semantic DNN RDMs as the independent variables, and each searchlight block RDM as the dependent variable. For each searchlight block, we selected the task type whose RDMs explained the highest variance uniquely as the function for that block. We used the nilearn (https://nilearn.github.io/index.html) library to plot and visualize the searchlight results.

## 4.6 Comparison of explained with explainable variance

To relate the variance of fMRI responses explained by a DNN to the total variance to be explained given the noisy nature of the fMRI data, we first calculated the lower and upper bounds of the noise ceiling as a measure of explainable variance and then compared cross-validated explained variance of each ROI by top-3 best predicting DNNs. In detail, the lower noise ceiling was estimated by fitting each individual subject RDMs as predictand with mean subject RDM of other subjects (N-1) as the predictor and calculating the $R^2$. The resulting subject-specific $R^2$ values were averaged across the N subjects. The upper noise ceiling was estimated in a

similar fashion while using mean subject RDMs of all the subjects (N) as the predictor. To calculate variance explained by the best predicting DNNs we fit the regression using cross validation in 2N folds (2 folds across conditions, N folds across subjects) where the regression was fit using the subject averaged RDMs of N-1 subjects and the fit was evaluated using $R^2$ on the left out subject and left out conditions. Finally, we then calculated the mean $R^2$ across 2N folds and divided it by the lower bound of the noise ceiling to obtain the ratio of the explainable variance explained by the DNNs.

### 4.7 Statistical testing

We applied nonparametric statistical tests to assess the statistical significance in a similar manner to a previous related study [64]. We assessed the significance of the $R^2$ through a permutation test by permuting the conditions randomly 10,000 times in either the neural ROI/ searchlight RDM or the DNN RDM. From the distribution obtained using these permutations, we calculated p-values as one-sided percentiles. We calculated the standard errors of these correlations by randomly resampling the conditions in the RDMs for 10,000 iterations. We used re-sampling without replacement by subsampling 90% (45 out of 50 conditions) of the conditions in the RDMs. We used an equivalent procedure for testing the statistical significance of the correlation difference and unique variance difference between different models.

For ROI analysis, we corrected the p-values for multiple comparisons by applying FDR correction with a threshold equal to 0.05. For searchlight analyses, we applied FDR correction to correct for the number of DNNs compared as well as to correct for the number of searchlights that had a significant noise ceiling.

We applied a two-sided t-test to assess the statistical significance of the cross-validated explained variance across N subjects. We corrected the p-values for multiple comparisons by applying FDR correction.

## Supporting information

**S1 Fig. Variance partitioning overview.** Given a set of multiple independent variables and dependent variables, multiple linear regression results in R-squared ($R^2$) that represents the proportion of the variance for a dependent variable that's explained by independent variables in a regression model. To find how 3 DNN RDMs together explain the variance of a given fMRI RDM we perform 7 multiple regression and illustrate unique and shared variance explained by models through a Venn diagram.
(TIFF)

**S2 Fig. Selecting task-specific DNN representation to compare with fMRI data. A)** Spearman's correlation of all DNN RDMs at a given layer of the encoder with other DNN RDMs computed at the same layer. We report the mean pairwise correlation of all 18 DNNs at different layers of the encoder. **B)** Spearman's correlation of all DNN RDMs at a given layer of the encoder with a randomly initialized model with the same architecture computed at the same layer. We report the mean correlation of all 18 DNNs with the randomly initialized DNN at different layers of the encoder. **C)** Spearman's correlation of all DNN RDMs at a given layer of the encoder with deeper layers (block4 and encoder output) of 2D DNNs. We report the mean correlation of the key layers of all 18 DNNs with deeper layers (block4 and encoder output) of 2D DNNs. **D)** Spearman's correlation between layers at different depths for DNNs corresponding to different task types. We report the mean correlation between different layers averaged across different DNNs of the same task type. **E)** Effect of adding all the key layers on unique and shared variance of fMRI RDMs from different ROIs as compared to selecting only

task-specific layers for variance partitioning analysis. We report the change in variance explained (variance change) for 7 variance partitions when all key layers were used for analysis as compared to selecting task-specific layers.
(TIFF)

**S3 Fig. $R^2$ ranking for 18 Taskonomy DNNs and random baseline in anatomical ROIs.** The bar plot shows the absolute total variance of each ROI RDM explained by task-specific layer RDMs of a given DNN. The asterisk denotes the significance of total variance ($p<0.05$, permutation test with 10,000 iterations, FDR-corrected across DNNs). The error bars show the standard deviation calculated by bootstrapping 90% of the conditions (10,000 iterations).
(TIFF)

**S4 Fig. $R^2$ ranking for 18 Taskonomy DNNs and random baseline in functionally localized ROIs.** The bar plot shows the absolute total variance of each ROI RDM explained by task-specific layer RDMs of a given DNN. The asterisk denotes the significance of total variance ($p<0.05$, permutation test with 10,000 iterations, FDR-corrected across DNNs). The error bars show the standard deviation calculated by bootstrapping 90% of the conditions (10,000 iterations).
(TIFF)

**S5 Fig. Effect of cross validation on variance explained ($R^2$). A)** Variance of each ROI explained by top-3 best predicting DNNs compared for different cross-validation settings (blue bars: no cross validation; orange bars: cross validation across subjects; green bars: cross validation across subjects and stimuli). The error bars show the 95% confidence interval calculated across N = 16 subjects. All the $R^2$ values are statistically significant ($p<0.05$, two-sided t-test, FDR-corrected across ROIs) **B)** Variance of each ROI explained by 1000 randomly generated RDMs compared for different cross-validation settings (blue bars: no cross validation; orange bars: cross validation across subjects; green bars: cross validation across subjects and stimuli). The error bars show the 95% confidence interval calculated across N = 16 subjects.
(TIFF)

**S1 Text. Selecting task-specific DNN representations.**
(DOCX)

## Author Contributions

**Conceptualization:** Kshitij Dwivedi, Radoslaw Martin Cichy, Gemma Roig.

**Data curation:** Michael F. Bonner.

**Formal analysis:** Kshitij Dwivedi, Michael F. Bonner.

**Funding acquisition:** Radoslaw Martin Cichy, Gemma Roig.

**Investigation:** Kshitij Dwivedi, Gemma Roig.

**Methodology:** Kshitij Dwivedi, Michael F. Bonner, Radoslaw Martin Cichy, Gemma Roig.

**Project administration:** Gemma Roig.

**Software:** Kshitij Dwivedi.

**Supervision:** Radoslaw Martin Cichy, Gemma Roig.

**Validation:** Kshitij Dwivedi.

**Visualization:** Kshitij Dwivedi.

**Writing – original draft:** Kshitij Dwivedi, Radoslaw Martin Cichy, Gemma Roig.

**Writing – review & editing:** Kshitij Dwivedi, Michael F. Bonner, Radoslaw Martin Cichy, Gemma Roig.

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
