## [Decision Letter · Decision Letter 0]

18 Apr 2021

Dear Mr. Dwivedi,

Thank you very much for submitting your manuscript "Unveiling functions of the visual cortex using task-specific deep neural networks" for consideration at PLOS Computational Biology.

As with all papers reviewed by the journal, your manuscript was reviewed by members of the editorial board and by several independent reviewers. In light of the reviews (below this email), we would like to invite the resubmission of a significantly-revised version that takes into account the reviewers' comments.

-

As you can see from the comments below the three reviewers were generally very positive about the analysis and modeling, but they also raise some concerns that can hopefully be addressed by a rewrite.

-

We cannot make any decision about publication until we have seen the revised manuscript and your response to the reviewers' comments. Your revised manuscript is also likely to be sent to reviewers for further evaluation.

Sincerely,

Ulrik R. Beierholm

Associate Editor

PLOS Computational Biology

Wolfgang Einhäuser

Deputy Editor

PLOS Computational Biology

Reviewer's Responses to Questions

**Comments to the Authors:**

Reviewer #1: In their manuscript, the authors compare task-specific CNNs to regions along the visual cortex to investigate their respective functionalities. The idea is that if a cortical region is involved in a specific task, it will show a stronger mapping to a CNN trained on a similar task compared to a CNN trained on a different task. The authors find that CNNs trained on low-level visual tasks better mapped to early brain regions, 3D-perceptual tasks to dorsal regions and semantic tasks regions in the ventral stream. I would further like to point out that the authors did a great job in setting up a website demonstrating their results with visualizations, etc. I really appreciated it! Overall, while the approach seems novel and promising and the data support coarse differences in functionality of different cortical regions, I have several concerns regarding the limitation of the tasks and neural data and the novelty of the findings per se.

Main comments:

1) My first comment concerns the generalizability and interpretation given the tasks and fMRI data are both limited to scene perception. Overall, I think the paper would be stronger if something could be added in terms of the functionality within cortical areas that are related to scene perception than trying to map these tasks/stimuli to the entire cortex. To give more details:

a. The authors focus on a set of CNNs trained on scene-related tasks based on the Taskonomy dataset. It is a great to have a set of different tasks for the same dataset, but these tasks have not been chosen to be biologically relevant and many of them seem less ecological relevant (e.g., normal, autoencoding, etc). The choice of tasks will affect the conclusion of what cortical area might be doing which task. The authors summarized those tasks as 2D, 3D and semantic. I agree that these coarse distinctions seem relevant for visual processing but it is hard to know what would happen if a different set of tasks was used for the analysis (e.g., subordinate, superordinate categorization, next-frame prediction, etc.), especially for areas which are not involved in scene processing. The authors discuss this limitation briefly but then don’t discuss the consequences for the interpretation of functionality in regions that are most likely not involved in scene perception. This needs to be added.

b. The fMRI dataset used in the manuscript was useful as it well fitted the CNNs’ training diet and was related to scene perception. However, we know that scene perception activates only a certain part of the human visual cortex (mainly RSC, OPA, PPA). Thus, this dataset and the fMRI experiment were optimized to achieve strong responses in early visual and scene-processing areas but most likely did not result in strong activation in many other areas (e.g., EBA, FFA, STS to name a few). This is reflected in the low noise ceiling achieved in these areas suggesting that there is not much signal to begin with. I think it is difficult to interpret the mapping between CNNs and areas that are barely responding to the stimuli. In my opinion, given this dataset, it would make more sense to focus on (ideally functionally-localized) areas related to scene processing and to test which task best explains voxels in PPA or OPA for example. There are still many open questions to a subdivision in PPA, the differences between OPA and PPA and others that could be investigated. If the authors could stand it, I think the paper would greatly benefit from looking at differences in those areas. In particular since the PPA, OPA, and RSC have all been functionally defined in the fMRI experiment the manuscript is based on (Bonner & Epstein, 2017). Without such an analysis, I am not sure what novel contribution the manuscript does on the functionality of the visual cortex, beyond introducing a novel method (It is of course fine to introduce a novel method but that would need restructuring and rephrasing the manuscript).

2) Related to the previous point, the noise ceiling in many of the cortical areas investigated is particularly low. Given a more diverse set of stimuli an R^2 of at least 0.1 in human IT can be achieved (e.g., Storrs et al., bioRxiv, 2020). The authors focus on the amount of explainable variance in the abstract and paper but given that the noise ceiling is so low this number is misleading (e.g., 60% of a highly noisy area is still low). This should be explicitly discussed.

3) The authors map the last two layers of the encoder of the CNNs to the fMRI data. They motivate this decision by the finding that representations in earlier stages of processing in the CNNs are not task-specific but shared across CNNs. However, as the authors describe in the manuscript, the visual cortex is build up in a hierarchy where later stages of processing build on earlier stages of processing. So it seems odd that to map an encoder output to early stages of processing in the visual cortex instead of earlier stages of the CNNs. How does the mapping of earlier stages (e.g., block 1/2) compare to their CNNs trained on 2D tasks? Also it would be good to see a correlation not just between the CNNs at different layers but also between layers. For example, is the correlation between early and late stages in the 2D-trained models higher than between models trained on 3D or semantic tasks?

Minor comments

1) It would be useful to add a figure in the manuscript showing the locations of the cortical areas investigated on an inflated or flattened cortex.

2) It was not clear to me how the representations from the last two layers in the encoder were build. Were the features concatenated? Or averaged? Please provide more details in the methods.

Reviewer #2: The authors provide a proof of principle that it is possible to use DNNs, trained on different tasks, to probe the function of brain areas. Using a collection of 18 DNNs, each trained with a different objective / loss function they show that three clusters of DNNs; one related to 2D perception, one related to 3D perception and one related to semantic information map on the expected regions; early visual cortex, the beginning of the dorsal pathway and the beginning of the ventral pathway.

That they findings from applying this approach are not surprising is a good thing because it validates the methods. The lack of clear additional findings, beyond the pale of what is roughly known of visual cortex is, from this perspective not a problem.

This is also a really nice illustration of the possibilities that the open science movement is providing because the paper combined the neural networks from one publication with the MRI data form another.

This is a novel approach and exciting approach. I do think that elements of the paper are overstated and some additional analysis might be sensible.

• This is one novel approach of mapping computational function to brain areas. It would be sensible to place this in a broader context of modern mapping research (for instance, researchers using genetic algorithms to maximally explain responses, or that use brain activity to train DNNs, for instance Seeliger et al, 2021). Given that the main novelty of this paper is in its approach it is sensible to discuss this in a broader framework.

• From another perspective there are also approaches in which a network is trained with multiple loss functions at the same time (for instance the UberNet of Kokkinos et al, 2016). This is from a different perspective another approach for mapping function to cortex. Again given that the main novelty of this paper is in its approach it is sensible to discuss this in a broader framework.

• Given that the network was trained on a different data-set than the data-set used for MRI it is important to describe its relationship. In terms of absolute R2 2D loss function DNNs are much better at explaining variance the than 3D / semantic loss function trained DNNs. Likely this is only due to the BOLD-MRI data-quality, but it would be good to hear that the variance in the MRI -dataset is not biased towards the 2D functions (and even if it is this is not a problem for this point of principle study, but still good to know).

• Looking at figure 2 it is clear that the noise level for areas beyond V3 (or even beyond V1) are low. This would suggest (see the preceding point) that the data-set is limited in terms of its variance. I have seen datasets where the noise ceiling drops less dramaticly between V1 and V3. Is this problematic for the analysis? At least I find it problematic that the researchers say they explain 60% of the explainable variance without pointing out that the data for these areas is very very noisy (and so without reporting the absolute r2).

• The researchers observe that the earlier layers are more correlated with each other and the later layers are more function specific. They therefore select the final 2 layers of the encoding network. However, given the many studies find the early visual cortex can be well explained by early layers of DNNs it would be sensible to include the first 2 layers as a baseline condition. I would expect that a part of the 2D responses are better explained by the first 2 layers, than be by the last two layers. Likely a (part of) V1 would fall out of analysis. Alternatively, even V1 is more complex than other analysis suggest.

• The researchers select from the 2D networks (7 loss functions) and 3D networks (8 loss functions) the 3 best performing networks. This is not possible with the semantic networks (only 3 loss functions come into play). Again, this is not a problem for the point of principle approach but is can result in effects on the r2 that are an effect of the selection procedure (vs the function. To unpack that, if loss functions are correlated they will share more units, and are therefore less capable of explaining variance than if they are uncorrelated (for instance see Scholte et al., 2018). Therefore, if the correlations between the loss functions are greater for the 2D functions (or 3D functions) they would be biased to have a higher r2. Vice versa, it is a relief to see that the semantic network is winning in VO1/VO2. Again this is not a problem for the point of principle nature of the paper but more on the correlation structure of the loss would be informative to interpret the results.

• It is not entirely clear to me what the novel functional insights for the 15 ROIs are. If they are really in the paper it would be sensible to outline this more directly.

• The authors take a stance on cognitive ontology. They refer to the framework of Marr in which function exists next to computation / implementation. However, an alternative is that there is a many to one, or even many to many mapping (see for instance: Francken, J. C., & Slors, M. (2014). From commonsense to science, and back: The use of cognitive concepts in neuroscience. Consciousness and cognition, 29, 248-258 or Klein, C. (2012). Cognitive ontology and region-versus network-oriented analyses. Philosophy of Science, 79(5), 952-960). Feel free to ignore the reference but using DNNs in this manner is sensible (or even more sensible) from this perspective.

Reviewer #3: In this manuscript, fMRI responses to 50 images of indoor scenes are compared against feature activations to these images in upper layers of a large number of DNNs obtained from the Taskonomy project, whereby each DNN was trained on a different task (with the same architecture). The ability of these DNNs to predict fMRI responses is evaluated via representational similarity analysis, in both a set of ROIs defined from a retinotopic atlas (visual regions V1 up to VO1/2 ventral and IPS) and in a search through the entire recorded brain volume. The authors find that DNNs pre-trained on different tasks (but with otherwise identical architectures) best predict different brain regions. In particular, representational similarity in lower visual regions is best predicted by DNNs trained on ‘low-level’ tasks like edge detection, ventral stream regions are best predicted by semantic tasks, and dorsal stream regions are best predicted by tasks that require representing 3D structure (and also semantic tasks). The authors conclude that the study ‘provides a comprehensive account of the function of human visual brain regions.’

Overall, I am sympathetic to the author’s goals here: I think the idea here is good and relevant, and I agree that this study is probably the largest so far in systematically comparing the effect that the task DNNs are trained on has on their ability to map fMRI responses in visual cortex. I also think that overall, the analyses are well executed (e.g., using variance partitioning on both the individual tasks and the task groups, with nice 3D visualizations), perhaps with the exception of the calculations going into the comparison of model explained variance with the maximal explainable variance, which may warrant some more substantiation (see comments below).

However, there are some weaknesses too. In particular, the claim above (“a comprehensive account”) is a bit of an overstatement, especially since very little interpretation is in fact offered of the obtained mappings. As pointed out by the authors, the mapping they find between the task groups and visual cortical regions ‘converges with’ a lot of prior research assigning functions to visual regions; it is indeed entirely consistent with our now a few decades old ‘standard’ model of visual processing existing of an early visual stage followed by a ventral/dorsal what/where pathway. So, a question that emerges is ‘do we actually learn something new here about how visual cortex is organized’? In this sense, the more fine-grained mapping of the individual tasks to individual ROIs offers potentially more interesting novel insights. However, I found these mappings somewhat difficult to make sense of, for two reasons: 1) results are obtained for one dataset with 50 images only, so may be idiosyncratic to the dataset and 2) there is very little explanation about the nature of the different tasks that are compared, and whether they may resemble tasks that the human visual system is also trying to solve.

Below, I provide some more detailed comments and suggestions that I hope will help the authors improve their manuscript.

MAIN COMMENTS

Methods:

As mentioned above, one limitation of this study is the fact that only one fMRI dataset with a relatively small set of images is tested, leaving open the question about to what extent the mapping of e.g., the ‘3D keypoints’ task with area V3b is a ‘real’ consistent finding or have something to do with the particular images used here. I realize it may not be easy to find another high-quality fMRI data set that uses indoor scenes (which the Taskonomy DNNs are optimized for), but (also given that the authors have themselves already published several papers on these data), I encourage them to consider testing the same models against another dataset, which I believe would greatly strengthen the paper (if it indeed shows consistent results as reported here). For example, perhaps there is a subset of images in the BOLD5000 dataset that are suitable for this purpose?

It would be really helpful if the paper would provide more insight into the nature of the different Taskonomy tasks, and if and how we should think of them as being ‘valid’ model tasks for what the brain might be doing. For example, autoencoding or denoising may not really be akin to the brain’s computational goals, whereas 2D edges may in fact be very much like the computational goal of, say, V1. I’m not saying the authors have to make strong claims, but it could be interesting to use these results to provide some insight into this question: what are ‘plausible’ functions of the brain regions studied here?

The claim that the models explain up to 60% of the explainable variance is impressive, especially given the fact that RSA studies typically report quite low correlations between model RDMs and fMRI RDMs. From section 4.6, I understand that the noise ceiling computation is an adaptation of the method described in the RSA toolbox paper (Nili et al, 2014) using explained variance rather than correlation. This makes sense to me. But it’s not clear to me that the calculation of the variance explained by the best predicting DNN here is valid in comparison; usually, the (group average of) single-subject correlations between model RDMs and fMRI RDMs are compared against the noise ceiling, but here these numbers are estimated by first fitting model RDMs to a subject-averaged RDM (of N-1 subject) and then testing those fitted regression weights against the RDM of the Nth subject (i.e., leave one out cross-validation). Isn’t this guaranteed to yield much higher R2 values (since you’re using the average to fit)? And how much does the 2N fold cross-validating across conditions affect the estimates, why is that included? In summary, I think that this new method needs to be explained, motivated and validated better, possibly through simulations using ground-truth RDMs, and/or how it compares to the ‘traditional’ RSA noise ceiling comparison using single-subject correlations only.

Results:

One thing that struck me as odd about the results is that in all the surface brain maps, the magenta ‘blobs’ (reflecting best predictions by semantic task-trained models) were always surrounded by an edge of green (best predictions by 3D task trained models). Why would this be? Could it be an artifact of the analysis?

Another thing that is striking is that in the dorsolateral view of the maps, there appears to be a strong lateralization with high correspondences with semantic-trained DNNs in the right hemisphere but none in the left hemisphere. Again, why would this be?

Discussion:

On line 378, it is suggested that this work highlights the feasibility of a ‘systematic approach that finds the functional roles of multiple brain regions […]’ Proof of principle is one thing, but do the results obtained so far generate any new hypotheses or predictions about the function of particular brain regions? Why (not)?

MINOR COMMENTS

According to the Methods, feature activations from the top 2 layers of each DNN are extracted and compared to the brain data, but it isn’t clear how these are then used in subsequent analysis. For example, is each layer used in the partitioning analysis, or do you pick the highest correlating one each time? Relatedly, I didn’t really follow the explanation offered on lines 464-469 about why the selection of the 2 top layers ‘ensured that differences in representation were due to the functions DNNs were optimized for’.

Methods line 573: For the ROI analysis, is the FDR correction applied only across models, or also across ROIs?

For Figure 1d, it is really hard to see any distinction between the blues, or between the different 3D or semantic tasks. In this sense, the figure is almost showing the same thing as Figure 3 where the RDMs are averaged within task groups prior to performing the searchlight. Would it be possible to provide additional maps showing only one task group, but with a more diverging color scale, for example?

It could be nice to show a couple of ROIs in more detail, for example how they represent the different tasks within a group (not just the top 3).

In Supp Section 2: The early layers of different DNNs show more similarity to a random init model then the later layers. Why is this the case? Do lower layers become less tuned than higher layers in training? Are they more shaped by the built-in architecture?

Discussion:

Given the large list of prior findings the results are consistent with, the findings here come across as purely confirmatory (and thus a bit boring). I’m wondering if a stronger claim can be made that this could be a proof of principle of not just that ‘it is possible to map the brain with DNNs’ but that the brain could - in theory - achieve different functions with the same canonical architecture and set of computations. In other words, it may not be necessary to have an entirely different type of CNN (with e.g. spatiotemporal units) to explain dorsal stream regions. Or does that go too far?

The authors seem to have missed a few relevant cites:

- Guclu & van Gerven (2017, NeuroImage) show a correspondence between dorsal visual regions and a DNN trained on action recognition (effectively also showing that dorsal regions are better explained by a different-trained CNN than object recognition)

- Lescroart & Gallant (2019, Neuron) showed that 3D information is a better predictor than 2D information is high-level scene regions through direct model comparison

Also, I think it should be highlighted that the same data has been used for comparison to CNNs (at the category-ROI level) in several previous studies.

LANGUAGE/TYPOS

Lines 213-214: the ‘suggesting a function related to’ here sounds a bit naïve, as if no-one every thought about the function of these ROIs before; ‘consistent with’ may be more appropriate here.

Lines 284-287: I found the distinction between ‘quantitative’ and ‘qualitative’ not very helpful here: to me it seems all these analyses are quantitative? The distinction is that one shows unique variance, and the other does not, so it just seems like a ‘stricter’ criterion perhaps?

Lines 288-291: The use of ‘group level’ here is a bit confusing, I think it refers to ‘groups of DNNs’ but since it is followed by a statement about ‘typical reporting in neuroimaging analysis’ this might be interpreted as reflecting subject-group level.

Lines 358: ‘revealed new functional insights for 15 ROIs’: again, what are these insights, and how new are they?

Lines 365: ‘both embody a theoretical hypothesis for explanation and predict brain responses well’ this sounds nice, but I don’t quite understand what is meant here.

Overall: in several places the word ‘the’ is missing before e.g., ‘model RDM’.

**Have the authors made all data and (if applicable) computational code underlying the findings in their manuscript fully available?**

Reviewer #2: Yes

Reviewer #3: Yes

PLOS authors have the option to publish the peer review history of their article (what does this mean?). If published, this will include your full peer review and any attached files.

Reviewer #1: No

Reviewer #2: **Yes: **H S Scholte

Reviewer #3: **Yes: **Iris Groen

**Have all data underlying the figures and results presented in the manuscript been provided?**

Reviewer #1: Yes
---

## [Decision Letter · Decision Letter 1]

11 Jul 2021

Dear Mr. Dwivedi,

We are pleased to inform you that your manuscript 'Unveiling functions of the visual cortex using task-specific deep neural networks' has been provisionally accepted for publication in PLOS Computational Biology.

Best regards,

Ulrik R. Beierholm

Associate Editor

PLOS Computational Biology

Wolfgang Einhäuser

Deputy Editor

PLOS Computational Biology

Reviewer's Responses to Questions

**Comments to the Authors:**

Reviewer #1: I thank the authors for their detailed and comprehensive revision and replies to the reviews. All my concerns have been addressed. I especially appreciated the new analysis comparing OPA and PPA. Finding differences between these two areas validates the approach and will certainly stimulate more research in this direction. I also thank the authors for pointing me to a confusion in Storrs et al. which indeed reported R instead of R^2. The new figure showing the correlation and noise ceiling per ROI is very helpful! Overall, the manuscript substantially improved and will certainly be interesting to a broad audience.

Reviewer #2: I thank the authors for responding to my review in a comprehensive way. In particular the extension in which the authors link the unique variance to functions of the DNNs is another highlight of the method.

Reviewer #3: The authors have addressed all my concerns. I think the addition of interpretation (of the models and their mappings to areas) really helped a lot in increasing the contribution of the paper!

**Have the authors made all data and (if applicable) computational code underlying the findings in their manuscript fully available?**

Reviewer #1: Yes

Reviewer #2: Yes

Reviewer #3: Yes

PLOS authors have the option to publish the peer review history of their article (what does this mean?). If published, this will include your full peer review and any attached files.

Reviewer #1: No

Reviewer #2: **Yes: **H.Steven Scholte

Reviewer #3: **Yes: **Iris Groen

---

## [Editor Report · Acceptance letter]

10 Aug 2021

PCOMPBIOL-D-21-00259R1 

Unveiling functions of the visual cortex using task-specific deep neural networks

Dear Dr Dwivedi,

I am pleased to inform you that your manuscript has been formally accepted for publication in PLOS Computational Biology. Your manuscript is now with our production department and you will be notified of the publication date in due course.

With kind regards,

Andrea Szabo
